# Transcriptome Profile Reveals Differences between Remote and Ischemic Myocardium after Acute Myocardial Infarction in a Swine Model

**DOI:** 10.3390/biology12030340

**Published:** 2023-02-21

**Authors:** María Pulido, María Ángeles de Pedro, Verónica Álvarez, Ana María Marchena, Virginia Blanco-Blázquez, Claudia Báez-Díaz, Verónica Crisóstomo, Javier G. Casado, Francisco Miguel Sánchez-Margallo, Esther López

**Affiliations:** 1Jesús Usón Minimally Invasive Surgery Centre, Carretera Nacional 521, Km 41.8, 10071 Cáceres, Spain; 2RICORS-TERAV Network, ISCIII, 28029 Madrid, Spain; 3CIBER de Enfermedades Cardiovasculares (CIBERCV), C. de Melchor Fernández Almagro, 3, 28029 Madrid, Spain; 4Immunology Unit, University of Extremadura, Campus Universitario, Av. de la Universidad, s/n, 10003 Cáceres, Spain; 5Institute of Molecular Pathology Biomarkers, University of Extremadura, 10003 Cáceres, Spain

**Keywords:** acute myocardial infarction, ischemic area, remote area, swine animal model, transcriptome profiling

## Abstract

**Simple Summary:**

Understanding the molecular basis of acute myocardial infarction is necessary to identify new therapeutic targets. In this work, a comparative transcriptome analysis was performed to identify differences in the expression profile between infarcted and remote areas of the myocardium in a porcine model of acute myocardial infarction.

**Abstract:**

Acute myocardial infarction (AMI) is the consequence of an acute interruption of myocardial blood flow delimiting an area with ischemic necrosis. The loss of cardiomyocytes initiates cardiac remodeling in the myocardium, leading to molecular changes in an attempt to recover myocardial function. The purpose of this study was to unravel the differences in the molecular profile between ischemic and remote myocardium after AMI in an experimental model. To mimic human myocardial infarction, healthy pigs were subjected to occlusion of the mid-left anterior descending coronary artery, and myocardial tissue was collected from ischemic and remote zones for omics techniques. Comparative transcriptome analysis of both areas was accurately validated by proteomic analysis, resulting in mitochondrion-related biological processes being the most impaired mechanisms in the infarcted area. Moreover, Immune system process-related genes were up-regulated in the remote tissue, mainly due to the increase of neutrophil migration in this area. These results provide valuable information regarding differentially expressed genes and their biological functions between ischemic and remote myocardium after AMI, which could be useful for establishing therapeutic targets for the development of new treatments.

## 1. Introduction

Myocardial infarction (MI) is a multifactorial disease and the most common cause of heart failure. This disease is associated with an ischemic injury caused by the partial or complete occlusion of coronary arteries. The ischemic area of the myocardium induces cardiomyocyte death, fibrosis, and deterioration in contractile function. This dysfunction is not only restricted to the infarcted area; anatomical alterations and molecular processes develop in the remote (non-infarcted) myocardium contributing to ventricle-wide remodeling [1]. These compensatory mechanisms include ventricular dilatation, hypertrophy, and fibrosis, playing an important role in the progress of heart failure.

The first consequence of ischemia is intracellular acidosis. Oxygen deprivation increases glycolysis to obtain energy, promoting the accumulation of lactate, protons, and NADH. Consequently, glycogen stores are depleted and are therefore insufficient to meet the energy demand [2]. Acidosis also disturbs the functioning of ion channels, resulting in an abnormal accumulation of intracellular calcium concentrations that contribute to impaired contractile function [3]. Indeed, local inflammatory reactions are triggered in the infarcted myocardium [4].

Inflammatory response and oxidative stress after MI are part of the healing and scar formation; however, exacerbated responses are related to a poor prognosis [5]. These mechanisms are activated in both the ischemic and remote cardiac tissue promoting profibrotic signaling cascades, impairing cardiac contractility and relaxation [6]. Therefore, cardiac remodeling affects the entire myocardium; nevertheless, the molecular differences between remote and infarcted myocardium remain largely unknown.

Thanks to the development of “omics” techniques in the last decade, the molecular pathways underlying MI are being elucidated. The knowledge of molecular mechanisms altered in MI is the basis to establish a therapeutic protocol. These techniques are on the rise due to the large amount of information provided. Recently, two studies have been published where spatial multiomics data reveal the molecular differences in myocardium after MI in human and mouse [7,8]. However, in the field of cardiology, although the porcine model is an excellent tool to evaluate the efficacy of different therapies, its use in “omics” studies has been limited by an incomplete annotation of the pig genome. Currently, pig databases have been completed, allowing deeper molecular analysis [9,10]. For example, a recent study has provided a genome resource of the major tissues and organs in pigs and their comparison to the human orthologs [11].

This study was aimed at determining the differences in the expression profiles between the infarcted and remote myocardial tissue regions of a MI in a relevant porcine animal model. Gene expression of cardiac tissue was determined by microarray analysis and proteomic validations, providing a biological sense of the different areas of the myocardium to which targeted therapies could be applied.

## 2. Materials and Methods

### 2.1. AMI Model Creation and Tissue Collection

All the study protocols were approved by the Ethics Committee on Animal Experiments of the Jesús Usón Minimally Invasive Surgery Centre, following the recommendations outlined by the local government (Junta de Extremadura, Spain), and the EU Directive 2010/63/EU of the European Parliament on the protection of animals used for scientific purposes. Moreover, this study is reported in accordance with the ARRIVE Guidelines [12].

Four young female Large White swine with an initial weight of 35–40 kg were used. All animals underwent a thorough clinical examination, and healthy animals were subjected to MI induction. Cardiac magnetic resonance (CMR) was performed 72 h after MI, and animals presenting with infarct size > 10% of the left ventricle and ejection fraction < 40% were included in the study (Appendix A).

Prior to infarct induction, oral amiodarone (400 mg) was given from 5 days before MI to 3 days after it. Acetylsalicylic acid (500 mg) was administered from 24 h before model creation until euthanasia. Furthermore, all pigs received prophylactic antibiotics for 5 days after MI induction.

MI induction was performed, as detailed in a previous study [13]. Briefly, anesthetized pigs were placed in dorsal decubitus to establish percutaneous access to a femoral artery. A coronary angioplasty balloon (Ryujin Plus, Terumo, Tokyo, Japan) of appropriate diameter was placed immediately distal to the origin of the first diagonal branch of the left anterior descending coronary artery and inflated for 90 min. During this occlusion period, possible arrhythmias or ventricular fibrillations were treated by manual chest compressions, 200 J biphasic defibrillation shocks (Zoll M Series Biphasic 200 J, Zoll Medical Corporation, Chelmsford, MA, USA), and pharmacological therapy when needed. Once the balloon had been deflated and removed, hemostasis of the arterial puncture site was performed using manual compression. The animals were kept under general anesthesia to treat possible malignant arrhythmias and were then recovered from anesthesia and carried to the animal housing facility.

Ten days after MI induction, CMR was performed (Appendix A) and animals were euthanized with an intravenous administration of 2 mmol/kg of KCl, applied under deep anesthesia.

After euthanasia, hearts were extracted and rinsed with PBS. The infarct volume was examined using 2,3,5-triphenyl tetrazolium chloride (TTC) staining, and myocardial tissue samples were collected from the mid-ventricular section in the anterior (infarcted area) and posterior (remote area) left ventricular wall. Tissue was minced and preserved in Allprotect Tissue Reagent (Qiagen, Hilden, Germany) at −20 °C.

### 2.2. Microarray Expression Analysis

Extracted RNA from myocardial tissue was processed using Affymetrix GeneChip WT Pico Reagent Kit (Thermo Fisher Scientific, Massachusetts, USA) based on the protocol designed and attached to the reagents. Data generation was assessed by scanning the GeneChip Porcine Gene 1.1 ST Array Plate in a Gene Titan Affymetrix microarray platform. Microarray data (CEL files) were normalized and annotated to genes according to the PorGene-1_1-st-v1 library using Transcriptomic Analysis Control software (Thermo Fisher Scientific). To estimate the significance and regulation of genes, differentially expressed genes (DEGs) were defined as those with absolute fold change (FC) > 2 and false discovery rate (FDR) < 0.05.

The microarray data have been deposited in the Gene Expression Omnibus (GEO) database with the GEO Submission GSE213657.

### 2.3. Proteomic Analysis

In order to validate transcriptomic data, protein expression was evaluated by a high-throughput multiplexed quantitative proteomic approach according to the previous protocol [14]. Briefly, protein extracts were digested, labeled using 8plex-iTRAQ reagents according to the manufacturer’s instructions (Waters Corporation, Milford, MA, USA), and analyzed by liquid chromatography-tandem mass spectrometry (LC-MS/MS). Peptide identification was performed by combining pig and human databases (UniProtKB/Swiss-ProtUniProtKB/Swiss-Prot 20147_02 07 Release), and proteins were annotated based on the Gene Ontology database [15]. Protein abundance changes were analyzed using a standardized variable, Zq, defined as the mean-corrected log2 (A/B) expressed in units of standard deviation at the protein level. Student t-test was used to compare Zq values from infarcted and remote areas, and the statistical significance was set at a value of *p* < 0.05.

### 2.4. Enrichment Analysis (Funtional Annotation)

DEGs identified by transcriptomic analysis and validated with proteomic results were subjected to bioinformatics analyses. The multi-omics data analysis tool, OmicsBean (http://www.omicsbean.cn/ accessed on 10 September 2022), was used to annotate levels of terms in biological process (BP), cellular component (CC), and molecular function (MF) based on Gene Ontology (GO) categories. Complementarily, a functional enrichment analysis was also carried out using g:Profiler (https://biit.cs.ut.ee/gprofiler/gost/ accessed on 10 September 2022) [16] with only annotated genes in pig, and the Benjamini–Hochberg method, applying significance threshold of 0.05. The biological pathway database Reactome (https://reactome.org/ accessed on 10 September 2022) [17] was utilized to define the biological pathways likely to be involved, depending on the DEGs identified. Lastly, protein–protein interaction (PPI) analysis was evaluated using Metascape (https://metascape.org/ accessed on 10 September 2022) [18] showing physical interactions in STRING (physical score > 0.132, the 33% percentile of the STRING physical distribution).

## 3. Results

### 3.1. Transcriptomic Analysis from Remote and Infarcted Myocardium Tissue

Changes in the gene expression levels between infarcted tissue and the remote zone of the myocardium (Figure 1A) were evaluated by transcriptomics analysis. The microarray technique identified 25,470 transcripts in every sample (14,430 identified with ID), whose level of expressions were compared between the different groups. This comparison analysis selected 916 transcripts (3.6% of the total identified genes) by applying the following filter criteria: (I) the Affymetrix ID is annotated in Gene Symbol; (II) fold change values between biogroups greater than 2 or less than −2; (III) FDR *p* value less than 0.05. A total of 535 transcripts (3.71% of total identified with ID) were down-regulated in infarcted tissue, and 381 (2.64% of the total identified with ID) were up-regulated. A total of 13,514 genes identified with ID did not meet criteria II and III (93.65%) (Figure 1B). All these transcripts are represented by their fold change and their FDR in a volcano plot (Figure 1C).

### 3.2. Proteomic Validation of Differentially Expressed Transcripts

Proteomic analysis was performed in remote and infarcted myocardium to validate transcriptomic results. High-throughput multiplexed quantitative proteomics allowed us to quantify a total of 2815 proteins. To compare this dataset with the 916 differentially expressed genes (DEGs) identified by microarrays, a Venn diagram is displayed (Figure 2A). Only 263 genes overlapped between the two datasets.

Correlation analysis of the expression changes between remote and infarcted myocardium for the 263 overlapped genes revealed a Pearson correlation coefficient of rp = 0.7225 between transcriptomics and proteomics. Most of the DEGs identified by transcriptomic were validated by proteomic quantification (233), identified by yellow and green dots, indicating that the expression trend is the same in both approaches. Moreover, six of these genes showed significant differences in protein expression levels between remote and infarcted myocardium (Appendix A). Unfortunately, 30 of these DEGs were not confirmed by proteomic results (Figure 2B).

### 3.3. Enrichment Analysis of Validated Genes

Enrichment analysis was performed between remote and infarcted tissue with those DEGs validated by proteomic results (Appendix A). According to this, the 233 genes were analyzed using the multi-omic data analysis online tool OmicsBean. Figure 3 shows an overview of Gene Ontology enrichment analysis, where the top ten more significant terms altered in infarcted myocardium are classified by the level they belong to and divided by the GO categories. Biological process categories are represented by specific terms such as *oxidation-reduction process* (GO:0055114), *extracellular matrix organization* (GO:0030198), or *cellular respiration* (GO:0045333) (Figure 3A). On the other hand, in the cellular component category, the major altered component is *mitochondrion* (GO:0005739) with 36% of DEGs in this term and the most significant *p* value (Figure 3B). Finally, 25% of DEGs are associated with *oxidoreductase activity* (GO:0016491), *NADH dehydrogenase (quinone) activity* (GO:0050136) being the term with the maximal annotated level in the molecular function category (Figure 3C).

In addition, DEGs validated by proteomics were divided by their expression trend (Figure 4A) and analyzed using g:Profiler online tool and Reactome data source. Enrichment analysis in up-regulated genes in the infarcted area revealed that these genes are related to processes such as *Extracellular matrix organization* (R-SSC-1474244), *Platelet activation, signaling, and aggregation* (R-SSC-76002), *Neutrophil degranulation* (R-SSC-6798695), *Collagen formation* (R-SSC-1474290), *Immune system* (R-SSC-168256) (the term with more up-regulated genes) (Figure 4B). On the other hand, down-regulated genes in infarcted myocardium are associated with terms including *The citric acid (TCA) cycle and respiratory electron transport* (R-SSC-1428517), *Respiratory electron transport* (R-SSC-611105), and *Striated Muscle Contraction* (R-SSC-390522) (Figure 4C).

### 3.4. Immune System Process in Infarcted Myocardium

Taking into account the relevance of the immune system in the myocardium after infarction, we analyzed the activation of pathways related to this term. For this purpose, up-regulated genes in infarcted myocardium and classified into the Immune system (R-SSC-168256) category were analyzed with the Reactome pathway online tool. This pathway is hierarchically divided into another three pathways: *Cytokine signaling in immune system* (R-HSA-1280215.5), with 6 entities found, *Innate immune system* (R-HSA-168249.8), with 16 entities found, and *Adaptive immune system* (R-HSA-1280218.5), with 9 entities found. In a more in-depth analysis, we can see that the most significant pathway is *Neutrophil degranulation* (R-HSA-6798695.2), with 12 of 19 genes associated with the Immune system pathway (Figure 5).

### 3.5. Protein–Protein Interaction Enrichment between Infarcted and Remote Myocardium

From the Metascape database, the protein–protein interaction network of the biological pathways was acquired, which relates the subset of proteins and their associated Reactome term following the molecular complex detection (MCODE) algorithm. The MCODE algorithm was applied to identify densely connected network components. In the whole screening process, eight clusters of MCODE with closely related functions were found and are displayed in different colors. The best-scoring terms by *p*-value of different MCODE are associated with *Complex I biogenesis* (R-HSA-6799198), *Respiratory electron transport* (R-HSA-611105), *Muscle contraction* (R-HSA-397014), and *Extracellular matrix organization* (R-HSA-1474244) (Figure 6).

## 4. Discussion

The literature describes the remote region of cardiac tissue after MI as a non-infarcted tissue that has not undergone oxygen deprivation but activates compensatory mechanisms that constitute a risk factor for heart failure development. In this sense, the remote region cannot be considered a healthy tissue because homeostasis in this area is also altered [6]. Knowledge of the molecular mechanisms underlying ischemic and non-ischemic tissue after acute myocardial infarction (AMI) is necessary to identify new therapeutic targets that will allow the development of treatment strategies focused on each myocardial area.

After AMI, apoptosis and necrosis mechanisms are activated in cardiomyocytes surrounding the infarcted area. The metabolic consequences of ischemia are reflected in remote areas of the myocardium where the surviving cardiomyocytes have to activate compensatory mechanisms to maintain cardiac functionality [19], involving multiple signaling pathways in different myocardium areas. However, the low percentage of differentially expressed transcripts between remote and infarcted areas shown in this study indicates that the altered mechanisms could be very similar and affect the entire myocardium.

To validate the transcriptomic results, proteomic analysis was performed. It is known that values of mRNA and protein abundance are not necessarily correlative because biological factors and technique limitations are important considerations that alter the central dogma of molecular biology [20]. In our dataset, the Pearson correlation coefficient (r_p_) between transcript and protein abundance was 0.7225, similar to the result obtained by *Popovic* et al. in single mammalian cells [21] and over other data in different organisms [20], which indicates that subsequent enrichment analyses are performed with accurate data.

Systems biology requires large datasets to draw meaningful conclusions from complex biological samples. Using our MI model, we attempted to reproduce a biological system to obtain valuable information. Our first approach was enrichment analysis with the differentially expressed genes between remote and ischemic myocardium. All GO categories indicate that the main mechanism altered between areas is mitochondrial respiration. In cardiomyocytes, oxygen deprivation activates processes related to impaired mitochondria, inhibiting the Krebs cycle, decreasing ATP production, and increasing ROS generation and intracellular accumulation of Na^+^ and Ca^2+^. Moreover, Ca^2+^ and ROS accumulation are increased during reperfusion, leading to the opening of the membrane permeability transition pore in the mitochondria and cardiomyocyte death [22]. Therefore, cardiomyocytes in the infarcted area suffer irreversible loss, and the infarcted myocardium cannot recover contractile function. However, surviving cardiomyocytes in the remote myocardium, which are also undergoing mitochondrial oxidative stress, become hypertrophied to compensate for the loss of function contributing to cardiac remodeling [23]. The combination of all these alterations triggers the functional impairment of the heart and makes the mitochondria a potential target for treatment.

Inflammatory and immune responses are mechanisms also activated after ischemia/reperfusion injury. The cells of the immune system and their secreted factors are known to play crucial roles in the initiation, progression, and resolution of inflammation [24]. In the inflammatory phase after MI, leukocytes are not only recruited to the site of the infarcted myocardium [25] but also to remote zones. These sections have been associated with the activation of pro-inflammatory pathways and infiltration of leukocytes [26]. In our analysis, the *Immune system* is the Reactome term with more genes up-regulated in infarcted tissue. These results are consistent with previous studies in which the accumulation of monocyte and macrophage in the remote myocardium is lower than in the infarct zone [27]. These differences could be due to the systemic inflammation caused by myocardial ischemia, where remote areas of the myocardium and remote organs express lower levels of proinflammatory cytokines and leukocyte infiltration [28].

In our protein–protein interaction enrichment analysis, most of the subsets of proteins are related to mitochondrion function. We found that expression of *mitochondria electron transport chain complex I* genes (*NDUFs*) are diminished in infarcted tissue. Similar results evaluating complex I activity in heart failure have been published elsewhere [29,30]. These findings suggest that the cardiomyocyte death in infarcted tissue may be due to the inactivation of complex I, which ultimately reduces oxygen respiration and ATP synthesis, as previously published [31]. In addition, complex I deficiency causes metabolic alterations, accumulating metabolites with consequent inflammation [31]. The low expression of glycolytic enzymes such as ENO3 and GAPDH reduces the production of glycolytic ATP, which is critical to maintaining contractility function. This mechanism is key in hibernating myocardium, where the overexpression of these enzymes guarantees the additional ATP production at low pressure of oxygen [32,33].

Some proteins related to hemostasis and platelet degranulation are dysregulated when remote and infarcted tissues are compared. *THBS1* is up-regulated in injured tissue in response to pressure overload, resulting in adverse cardiac remodeling [34]. In mice, the treatment with AngII increased *THBS1* expression in the heart, which in turn increased collagen deposition during cardiac remodeling. Moreover, AngII stimulates cardiac hypertrophy induced by calcium overload through the impaired function of ATP2a2, an ATPase that pumps calcium into the sarcoplasmic reticulum lumen, which is down-regulated in infarcted tissue [35]. On the other hand, our results showed an increase in the expression of *ANXA5*, similar to that in human patients with myocardial infarction in which cell death in the infarct area was marked by annexin-V after reperfusion [36].

Other evidence confirming that mitochondria are the main compartment affected in ischemic tissue compared to remote is the interaction proteins related to the citric acid (TCA) cycle. The down-regulated genes *ACO2* and *IDH2* participate in the citrate and isocitrate metabolic pathways in the TCA cycle in the mitochondrial matrix and the cytosol (COMPLEX I) [37]. MI patients have a higher incidence of *IDH2* gene variations [38], and in the mouse model of I/R, mitochondria isolated from ischemic tissue showed a diminished expression of *ACO2* [39].

In terms of cardiac structural proteins that contribute to muscle contraction, several genes differ in the infarcted myocardium. Sarcomeric proteins such as actin, myosin, tropomyosin, or myosin-binding proteins, encoded by the *ACTN2*, *TPM1*, *MYL3*, and *MYBPC3* genes, were down-regulated in ischemic tissue. While remote tissue maintains an impaired contractility function, the ischemic area cannot recover this functionality. All of these genes have been evaluated in the interventricular septal myocardium of hypertrophic cardiomyopathy human patients showing different expression patterns when compared to control donors [40]. This may suggest that contractility is impairing the entire myocardium, although the degree of impairment is higher in the ischemic area.

In the proliferative phase of MI, myofibroblasts accumulate in the infarcted area and produce large amounts of extracellular matrix proteins. This collagen-based matrix is composed of other proteins, such as thrombospondin 1 or SPACRC, which promotes matrix organization, resulting in a mature scar [41]. Our results showed an increase in the expression of these proteins in the ischemic area, indicating that the proliferative phase begins one week after MI induction.

The use of healthy animals in these procedures is one of the limitations of this study. Therefore, in order to put the results in context, it should be taken into account that humans are generally diabetic, hypertensive, obese, etc., and these conditions could alter the obtained results in healthy pigs.

## 5. Conclusions

In conclusion, the aim of this study was to investigate differences in gene expression between remote and infarcted areas from infarcted myocardium. The combined use of transcriptomic and validation with proteomic analysis has been shown to be an effective method to establish the main differences between infarcted and remote tissues. As a result, the main impaired pathways were mitochondria respiration and extracellular matrix organization; however, the expression pattern in the immune system is related to both areas due to systemic inflammation triggered by myocardial infarction. Our study provided valuable information that could contribute to a deeper understanding of the molecular mechanism activated in the entire myocardium after MI and establish therapeutic targets in targeted treatment in the myocardial area of interest.

## Figures and Tables

**Figure 1 biology-12-00340-f001:**
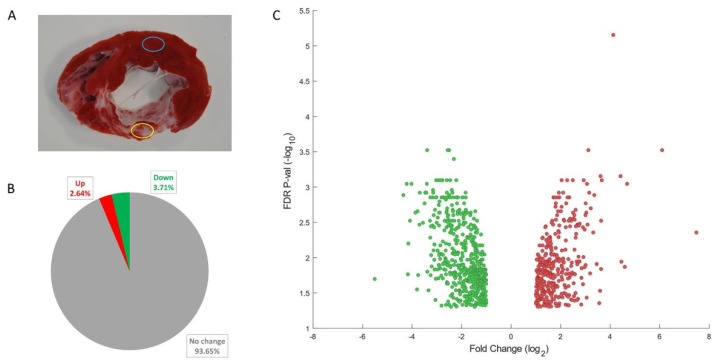
Transcriptomic results. Representative TTC-stained heart slice obtained from an animal belonging to the study. Sampling was performed from the ischemic (yellow circle) and remote (blue circle) areas (**A**). Pie chart represents the 14,430 transcripts identified by microarrays and assigned with ID, where the green slice represents the proportional value of down-regulated genes, the red slice represents the up-regulated genes, and the grey slice represents no changed genes (**B**). DEGs are visualized in a volcano plot according to FDR p-value (-log10) and fold change (log2). Green dots represent down-regulated genes and red dots, up-regulated genes (**C**). DEGs: differentially expressed genes. FDR *p*-value < 0.05. Fold change: >2 or <−2.

**Figure 2 biology-12-00340-f002:**
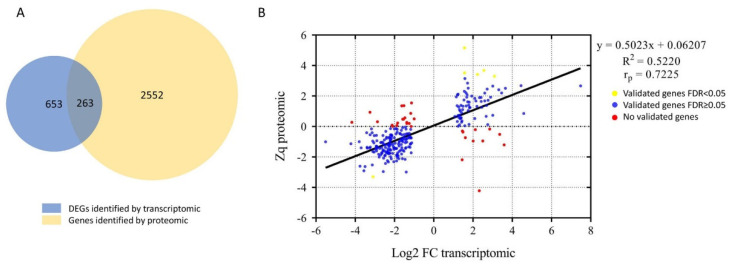
Proteomic validation. Venn diagram shows the intersection between DEGs identified by transcriptomic (yellow circle) and genes identified by proteomic (blue circle) (**A**). The scatter plot shows the correlation analysis of the expression changes through both transcriptomics and proteomics for the Venn diagram intersection data, including linear regression analysis. Genes that were significantly regulated (FDR < 0.05) in both RNA and protein are highlighted in yellow, validated genes that were not significantly regulated (FDR > 0.05) in protein are highlighted in blue, and not validated by protein abundance are highlighted in red (**B**). DEGs: differentially expressed genes. R^2^: coefficient of determination. r_p_: Pearson correlation coefficient.

**Figure 3 biology-12-00340-f003:**
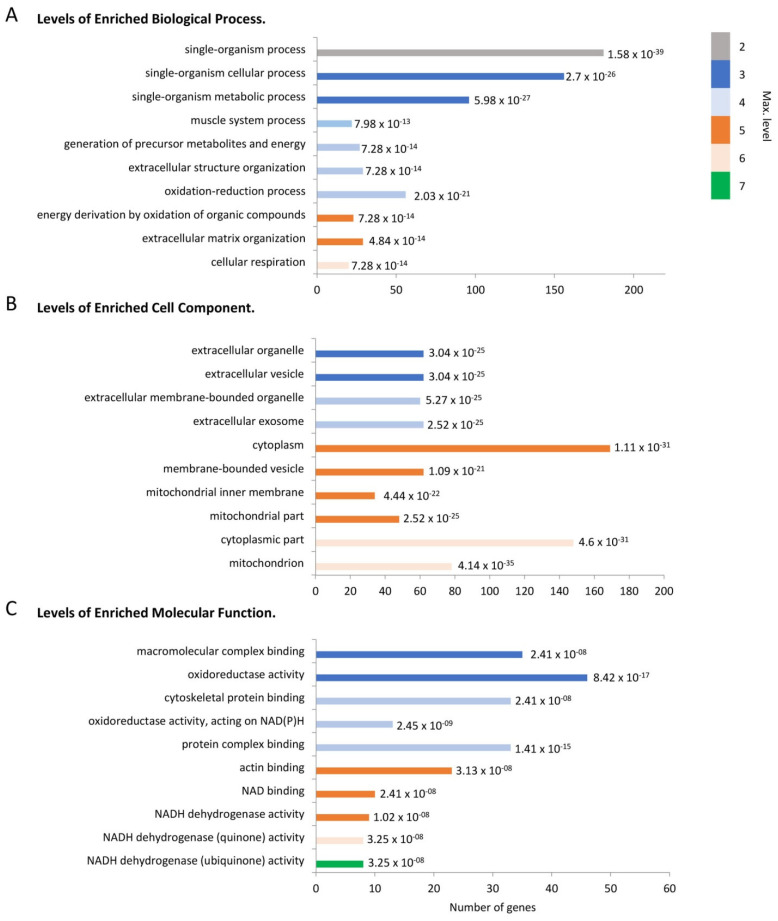
Enrichment analysis of validated genes. Bar charts show enrichment results using only validated DEGs by proteomics. Levels of enriched biological process (**A**), enriched cell component (**B**), and enriched molecular function (**C**) are expressed in percent of genes and divided by max level. DEGs: differentially expressed genes. Max Level: maximal annotated level of this term in the GO graph (tree).

**Figure 4 biology-12-00340-f004:**
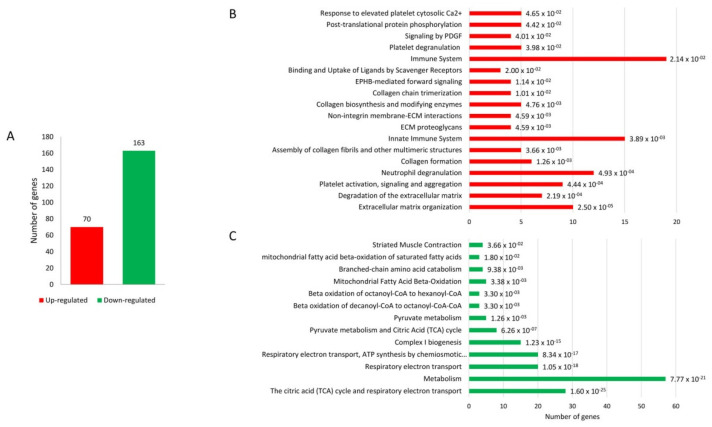
DEGs up- and down-regulated. Number of DEGs validated by proteomics and divided in up- and down-regulated are shown in bar graph (**A**). Enrichment analysis of both datasets were performed using g:Profiler tool and Reactome data source. The green bar graph represents number of down-regulated genes in Reactome terms (**B**), and the red graph, the number of up-regulated genes in Reactome terms (**C**).

**Figure 5 biology-12-00340-f005:**
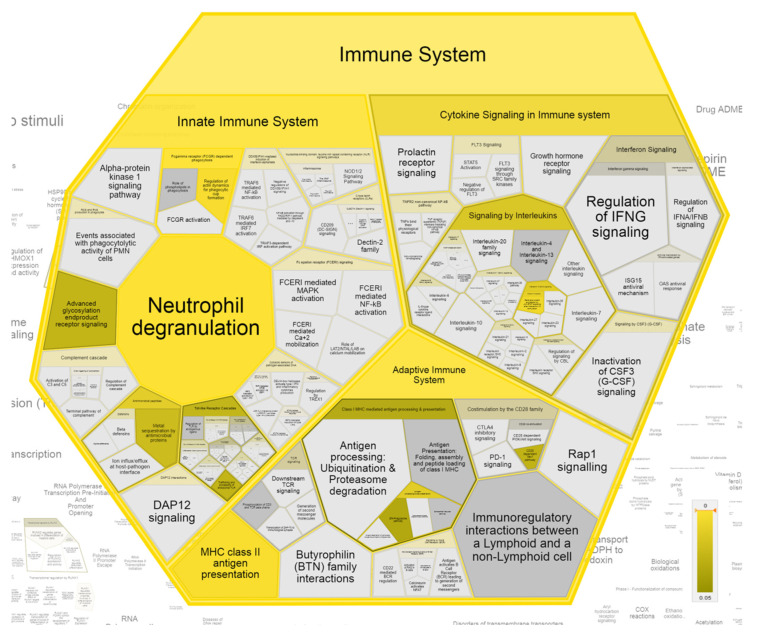
DEGs in the immune system process. Voronoi pathway visualization (Reacfoam) for the identified proteins in the *Immune system* (R-SSC-168256) term. The yellow color code denotes over-representation of that pathway in our input dataset. Light grey signifies pathways which are not significantly over-represented.

**Figure 6 biology-12-00340-f006:**
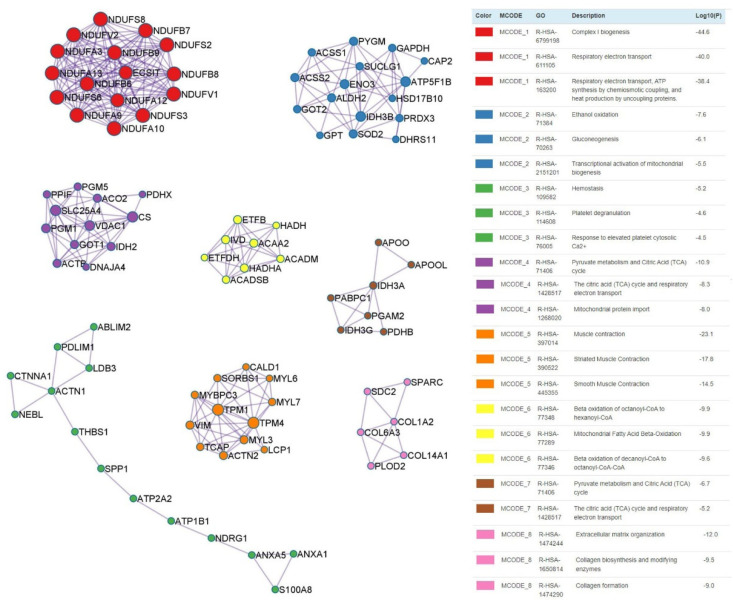
Protein–protein Interaction Enrichment Analysis. Protein–protein interaction analysis of validated DEGs by using the molecular complex detection (MCODE). Each network represents different clusters where the temporal dynamic clustering analysis MCODE algorithm was applied. Reactome enrichment analysis was applied to each MCODE network. Each node represents proteins, and the edge represents the specific interaction between proteins. DEGs: differentially expressed genes.

## Data Availability

The data presented in this study are openly available in the Gene Expression Omnibus (GEO) database with the GEO Submission GSE213657.

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
