# Peer review of "Transcriptome Profile Reveals Differences between Remote and Ischemic Myocardium after Acute Myocardial Infarction in a Swine Model"

_biology, 2023, doi:10.3390/biology12030340_

Round 1
Reviewer 1 Report
The work entitled “Transcriptomic profiles reveal differences between remote and ischemic myocardium after acute myocardial infarction in swine model” by Maria Polido and colleagues reaches out to analyse the protein expression in acute myocardial infarction as compared to remote areas. The study utilizes the porcine model, which is attractive because it resembles size and anatomy of the human heart quite closely. The transcriptomic data were validated by proteomics. This approach potentially eliminates some differentially expressed genes from the results because the transcriptome analysis is potentially more sensitive than proteomics. As a strength of this combined approach the resulting set of differentially expressed proteins is highly significant.
This is a robust study to generate a set of data of differentially expressed genes in the acute cardiac infarction and could be of use as a reference for future projects. As a limitation, novel insights are limited in this approach as it appears mostly descriptive.
Author Response
We agree with Reviewer 1 regarding the limitations of the study. This work is mainly descriptive because we are focused on the comparative description between remote and infarcted myocardium. Our methodological approach, using a clinically relevant animal model under controlled conditions has been really helpful to identify potential therapeutic targets in the treatment of myocardial infarction.
Moreover, the multiple enrichment analysis provides an overview of those biological processes altered in the infarcted myocardium which may help to define present and future therapies.
Reviewer 2 Report
The axis labels in Figure 1C should be modified to improve clarity and precision.
It is unclear why the authors chose to use a standard of physical score > 0.132 in STRING.
A scatter plot including all expressed genes may provide useful information to validate the consistency of the transcriptomics and proteomics data.
The Results section appears to lack depth and detail, providing more of a description rather than a thorough analysis of the findings.
Author Response
Point 1. The axis labels in Figure 1C should be modified to improve clarity and precision.
Respond 1. Following the suggestions of the Reviewer, Figure 1C has been modified accordingly.
Point 2. It is unclear why the authors chose to use a standard of physical score > 0.132 in STRING.
Respond 2. In order to clarify this question, we have included the following sentence in the text: the 33% percentile of the STRING physical distribution (Line 156).
Metascape web (https://metascape.org/blog/?p=219) explain in detail why is correct to choose a physical score > 0.132. Following, we provide the exhaustive explanation.
STRING provides a probabilistic framework to assign a confidence score for each PPI pair, by assuming all evidences are independent. We therefore can assign both a physical score and a combined score for its data record. But how to assign a score to data not captured in STRING, so they can be combined?
First, for those PPI pairs that are already included in STRING, we check their STRING physical scores. The figure below shows the physical score distribution of BioGrid physical subset, BioGrid functional subset, OmniPath, InWebDB and STRING physical subset itself using human data. Notice these are accumulative curves for their score distributions. We can see about 50% of the PPI data in OmniPath and InWeb_DB have a physical score > 0.9, i.e., these two data source indeed are of high quality even by their STRING physical scores. Then BioGrid physical subset has better quality than its functional subset and STRING subset has the lowest quality. I.e., in terms of data source quality, we can say OmniPath > InWeb_DB > BioGrid (Physical) > BioGrid (Functional) > STRING (Physical), in line with we expected.
Now since we cannot assign individual STRING scores to those pairs that are not already in STRING, we can only assume all data in non-STRING data sources share the same STRING physical score. We subjectively choose the score corresponds to ~33% percentile (1/3 of the height in the accumulative curve) of the above distribution. That is we set OmniPath, InWeb_IM and BioGrid (Physical) a STRING physcial score of 0.537, 0.356, 0.260, respectively. Then we take the 33% percentile of the STRING physical distribution itself, 0.132, as the cutoff. Therefore, all physical interactions with STRING score > 0.132 are consider a reliable subset, which we call “Physical (Core)”. “Physical (Core)” include all of OmniPath, InWeb_DB, BioGrid Physical and 2/3 of STRING Physical. Then all physical interactions, regardless of their STRING scores are included in the “Physical (All)” dataset.
Point 3. A scatter plot including all expressed genes may provide useful information to validate the consistency of the transcriptomics and proteomics data.
Respond 3. Taking into account Reviewer's suggestion, we have referenced table S1 (Line 192) in the Results section where we describe the results of the scatter plot figure. This table includes all DEGs validated by proteomics including their gene names, transcriptomic fold change data, proteomic Zq data (the mean-corrected log2−ratio expressed in units of standard deviation at the protein level), and their respective FDR P-vale. We agree that this information could be useful to readers.
Point 4. The Results section appears to lack depth and detail, providing more of a description rather than a thorough analysis of the findings.
Respond 4. Thank you for this appreciation. The Results section has been revised and we have included several details highlighted in yellow that clarify our results. In addition, CMR analysis has been added as a supplementary figure, indicating infarct size and location, which could be helpful for better interpretation of our model.
Reviewer 3 Report
In their manuscript entitled “Transcriptomic profiles reveal differences between remote and ischemic myocardium after acute myocardial infarction in swine model”, Pulido et al. present data on a transcriptomic and proteomic analysis of the infarct and remote areas of the heart in a swine model of cardiac ischemia. They conclude that mitochondrion-related and immune-related processes differ the most between the two locations in the heart. The interesting aspect of this study is that proteomic and transcriptomic techniques were combined to analyze the gene/protein expression in the porcine infarct model, which is generally considered to be more predictive to the human situation than rat or mouse models. While reading the manuscript, some questions came up:
1. The timeline of the experiment is not very clear. At 72 hours post-MI a CMR was performed (line 93-94) and antibiotics were administered till day 5 post-MI (line 98-99), but I could not find when the animals were euthanized.
2. It would be helpful to include the results of the CMR analysis in the manuscript. In my experience, the size and location of the infarct are important in determining the impact on cardiac function. Could you also include the staining method used in Fig. 2a (I presume it is triphenyltetrazolium but I could not find it in the manuscript)?
3. Line 48 and 52: the authors use the word scar to describe the matrix deposition in the infarct area, but also in the remote area. I would prefer the term ‘fibrosis’ for the latter as it will typically surround the surviving cardiomyocytes rather than replacing them.
4. line 183: trendy should read trend
5. line 280-281: the Pearson correlation coefficient should read 0.7225 according to Fig 2B.
Author Response
Point 1. The timeline of the experiment is not very clear. At 72 hours post-MI a CMR was performed (lines 93-94) and antibiotics were administered till day 5 post-MI (lines 98-99), but I could not find when the animals were euthanized.
Respond 1. We agree that the time of euthanasia is not specified in the text. Therefore, we have added some lines (lines 111-113) to clarify the model's timeline.
Point 2. It would be helpful to include the results of the CMR analysis in the manuscript. In my experience, the size and location of the infarct are important in determining the impact on cardiac function. Could you also include the staining method used in Fig. 2a (I presume it is triphenyltetrazolium but I could not find it in the manuscript)?
Respond 2. We fully agree with the Reviewer on this point. The CMR data analysis has been included as supplementary material. We have performed a new supplementary figure showing representative images of CMR and a table with cardiac function parameters of all studied animals. In the images, we can see the localization of the infarcted area in the anteroseptal myocardium. Therefore, cardiac function is compromised.
Moreover, the staining method performed in Figure 1A has also been added in the Material and Methods section (line 115) and in Figure Legend 1 (line 172).
Point 3. Line 48 and 52: the authors use the word scar to describe the matrix deposition in the infarct area, but also in the remote area. I would prefer the term ‘fibrosis’ for the latter as it will typically surround the surviving cardiomyocytes rather than replacing them.
Respond 3. Following the suggestions of the Reviewer, the term “scar” has been substituted by “fibrosis”
Point 4. line 183: trendy should read trend
Respond 4. Sorry about the mistake. "Trendy" has been replaced by "trend" in the text.
Point 5. line 280-281: the Pearson correlation coefficient should read 0.7225 according to Fig 2B.
Respond 5. Thank you for this observation. The error has been corrected in the text.
Round 2
Reviewer 2 Report
Most of the results appear satisfactory. However, Some suggestions to improve the presentation of the results are as follows:
In Fig 1C, use integers or .5 separation for the horizontal coordinates.
Remove the label "FDR P-val vs Fold Change" from Fig 1C.
Reconsider the color scheme for most of the plots.
In Figure 6, make the protein names clearer.
Author Response
Point 1. In Fig 1C, use integers or .5 separation for the horizontal coordinates.
Remove the label "FDR P-val vs Fold Change" from Fig 1C.
Respond 1. Thank you to Reviewer 2 for this appreciation. Fig. 1C has been modified accordingly.
Point 2. Reconsider the color scheme for most of the plots.
Respond 2. Thank you very much for the comment.
The color scheme used is the default scheme used by the analysis software as software used in this study (Transcriptome Analysis Console). This same scheme, in which up-regulated genes are represented in red and down-regulated genes in green, has been observed in numerous papers. Furthermore, these data are correctly described in the legend and in the figure caption.
However, we have modified figure 2B to avoid misinterpretation of the results.
Point 3. In Figure 6, make the protein names clearer.
Respond 3. Figure 6 has been modified following the suggestions of Reviewer 2